# Broad functional profiling of fission yeast proteins using phenomics and machine learning

**María Rodríguez-López[1†], Nicola Bordin[2†], Jon Lees[2,3], Harry Scholes[2], Shaimaa Hassan[1,4], Quentin Saintain[1], Stephan Kamrad[1], Christine Orengo[2]\*, Jürg Bähler[1]\***

[1]University College London, Institute of Healthy Ageing and Department of Genetics, Evolution & Environment, London, United Kingdom; [2]University College London, Institute of Structural and Molecular Biology, London, United Kingdom; [3]University of Bristol, Bristol, United Kingdom; [4]Helwan University, Faculty of Pharmacy, Cairo, Egypt

**\*For correspondence:**
c.orengo@ucl.ac.uk (CO);
j.bahler@ucl.ac.uk (JB)

[†]These authors contributed equally to this work

**Competing interest:** The authors declare that no competing interests exist.

## Abstract

Many proteins remain poorly characterized even in well-studied organisms, presenting a bottleneck for research. We applied phenomics and machine-learning approaches with *Schizosaccharomyces pombe* for broad cues on protein functions. We assayed colony-growth phenotypes to measure the fitness of deletion mutants for 3509 non-essential genes in 131 conditions with different nutrients, drugs, and stresses. These analyses exposed phenotypes for 3492 mutants, including 124 mutants of 'priority unstudied' proteins conserved in humans, providing varied functional clues. For example, over 900 proteins were newly implicated in the resistance to oxidative stress. Phenotype-correlation networks suggested roles for poorly characterized proteins through 'guilt by association' with known proteins. For complementary functional insights, we predicted Gene Ontology (GO) terms using machine learning methods exploiting protein-network and protein-homology data (NET-FF). We obtained 56,594 high-scoring GO predictions, of which 22,060 also featured high information content. Our phenotype-correlation data and NET-FF predictions showed a strong concordance with existing PomBase GO annotations and protein networks, with integrated analyses revealing 1675 novel GO predictions for 783 genes, including 47 predictions for 23 priority unstudied proteins. Experimental validation identified new proteins involved in cellular aging, showing that these predictions and phenomics data provide a rich resource to uncover new protein functions.

## eLife assessment

This **important** study combines extensive phenotyping of genome-wide deletion mutants and machine learning-based prediction to generate a large scale resource for understanding the functions of thousands of fission yeast protein-coding genes. This resource is supported by **convincing** phenotyping data and state-of-the-art bioinformatic analyses and will be of interest to many geneticists.

## Introduction

Most biomedical research focuses on genes that are already well-studied, a situation which has changed little in the past 40 years of biochemical, cell biological, and genetic investigations (*Kustatscher et al., 2022*; *Edwards et al., 2011*; *Stoeger et al., 2018*; *Pfeiffer and Hoffmann, 2007*; *Su and Hogenesch, 2007*; *Haynes et al., 2018*; *Oprea et al., 2018*; *Wood et al., 2019*; *Sinha et al., 2018*).

The cellular functions of many genes thus remain poorly characterized or unknown, even with the availability of whole-genome sequences (*Gates et al., 2021*). This disparity is reflected in a strong bias in citations and publications which has remained constant over the past two decades (*Edwards et al., 2011*; *Stoeger et al., 2018*). A synthetic bacterium with a minimal genome contains 473 genes, defining a bare essential for life *Hutchison et al., 2016*; remarkably, 149 (~32%) of these most vital genes have unknown cellular roles. Moreover, a recent proteome survey across the evolutionary range finds that 38.4% of the identified proteins are not associated with any biological process, including 22.9% of the 100 most abundant proteins of each species (*Müller et al., 2020*). These examples highlight how much of the 'dark proteome' awaits discovery and the need to characterize gene function in an unbiased manner. To fully understand cells, we ought to know what all parts do and how they contribute to biological systems and disease.

The fission yeast, *Schizosaccharomyces pombe*, provides a powerful platform to interrogate eukaryotic gene function owing to its relative simplicity, well-annotated genome, deletion-mutant libraries for high-throughput assays, and genetic tractability under tightly controlled conditions (*Rallis and Bähler, 2016*). The *S. pombe* genome annotation contains 5134 protein-coding genes, 3624 (70.6%) of which are conserved in metazoa. Unlike budding yeast, fission yeast does not undergo any genome duplication, so there is less gene redundancy and mutations are more likely to result in phenotypes (*Wood et al., 2002*). Moreover, unlike in mammalian cells, ~90% of the genes are expressed under standard growth conditions (*Marguerat et al., 2012*), facilitating analyses of their functions at the organism level. Fission yeast resembles mammalian cells in many respects (e.g. symmetrical cell division, chromatin and RNAi pathways, centromeres, and replication origins), and thus provides a complementary model system to the more widely studied budding yeast (*Hoffman et al., 2015*). As in other model organisms (*McGary et al., 2010*), genes are being associated with cellular functions and phenotypes at higher rates in fission yeast than they are in humans. Fission yeast researchers, supported by the model organism database PomBase (*Harris et al., 2021*; *Lock et al., 2019*), have experimentally characterized 2498 proteins in publications, and the biological functions of 1916 additional proteins can be reliably inferred from orthologs in other organisms (status March 2023). Systematic genetic screens have been published for several, mostly well-studied conditions, including DNA damage (*Deshpande et al., 2009*), caffeine tolerance (*Calvo et al., 2009*), catalase expression (*García et al., 2016*), mycelial development (*Dodgson et al., 2009*), aneuploid viability (*Tange et al., 2012*), autophagy (*Sun et al., 2013*), heavy-metal tolerance (*Guo et al., 2016*), cell-cycle progression and cell shape (*Hayles et al., 2013*), mitotic competence (*Sajiki et al., 2018*), TORC1 inhibition (*Rodríguez López et al., 2020*; *Lie et al., 2018*; *Rallis et al., 2014*), respiratory growth (*Malecki and Bähler, 2016*; *Malecki et al., 2016*; *Zuin et al., 2008*), and chronological lifespan (*Sideri et al., 2015*; *Romila et al., 2021*). Nevertheless, the number of entirely unknown proteins is hardly decreasing over time; so without fresh approaches and resources directed toward unknown proteins, it will take dozens of years to uncover their functions (*Wood et al., 2019*). Empirical data show that genes that are linked with biological processes and functions in model organisms become characterized much more readily in other organisms (*Stoeger et al., 2018*; *Haynes et al., 2018*; *Wood et al., 2019*; *Hu et al., 2017*; *Mungall et al., 2017*). Thus, initial dedicated efforts to link unstudied proteins with cellular functions and biological processes are key to trigger deeper follow-on analyses of these proteins.

The concept of protein function is somewhat imprecise and hierarchical, involving several layers of detail (*Szklarczyk et al., 2015*). PomBase labels *S. pombe* genes as 'unknown' if they cannot be associated with any informative, high-level GO Biological-Process terms (*Carbon et al., 2021*), applying strict criteria based on functional assays for mutant phenotypes, protein localization, and interactions in *S. pombe*, or on corresponding functional assays in other organisms to infer the function (*Wood et al., 2019*). *S. pombe* encodes 641 'unknown' genes (PomBase, status March 2023). There are of course many more poorly characterized proteins for which little meaningful information is available because the associated annotations are either incorrect or too general to be informative. Among these 641 unknown proteins, many are apparently found only in the fission yeast clade, but 380 are more widely conserved. Among these conserved proteins, 135 'priority unstudied' proteins show clear human orthologs, and our analysis using Orthologous Matrix (*Altenhoff et al., 2019*) reveals broad and strong conservation profiles across 100 metazoans. These priority unstudied proteins have not been directly studied in any organism but can be assumed to have pertinent biological roles conserved

over 1000 million years of evolution. Notably, 49 (~36%) of the 135 unstudied proteins conserved from fission yeast to humans are not conserved in budding yeast, the best characterized eukaryote.

Initial efforts to learn about unknown genes need to be broad and exploratory, because hypothesis-driven research cannot target proteins for which nothing is known. Many genes may remain unknown because they are not required under benign laboratory conditions. Phenomics seeks to uncover gene function by rigorously identifying all phenotypes associated with gene mutants under many different conditions (*Rallis and Bähler, 2016*; *Brochado and Typas, 2013*). Microbes like yeast are particularly suitable for such high-throughput phenotyping. For example, while only 34% of all budding yeast deletion mutants display growth phenotypes under a standard condition, 97% show phenotypes in specific chemical or environmental perturbations (*Hillenmeyer et al., 2008*). Such broad genetic screens provide a basis to understand the function of unstudied genes. Besides offering direct functional clues about biological processes being affected by specific genes, they also give rich phenotypic signatures reflecting systemic responses to genetic perturbations in different conditions. Clustering and network analyses can point to the functions of unstudied genes that show similar phenotypic signatures to known genes, a principle called 'guilt by association' (*Ryan et al., 2013*; *Wiwie et al., 2015*).

Besides phenomics, advances in machine learning now provide potent opportunities for unbiased functional predictions based on available data associated with genes (*Huttenhower et al., 2009*; *Libbrecht and Noble, 2015*; *Angermueller et al., 2016*). Over the past decade, various approaches have been independently benchmarked to determine the most useful strategies. Recent evaluations highlight the importance of protein-sequence homology as a key component of successful methods (*Zhou et al., 2019a*; *You et al., 2018*). Future trends are likely to see more important contributions from deep-learning feature extraction on protein sequences (*Sanderson et al., 2021*). A particularly successful homology-based strategy in the last round of the Critical Assessment of Functional Annotations, CAFA4, exploits CATH protein functional families (FunFams) and has consistently been highly ranked for precision in Gene Ontology (GO) Molecular Function and Biological Process prediction CAFA1 (*Radivojac et al., 2013*), CAFA2 (*Jiang et al., 2016*), CAFA3 (*Zhou et al., 2019a*) and CAFA4 (ISMB2020, personal communication from organizers). However, homology-based data are not the only useful source for functional prediction, and combining other data types with machine learning can help to maximize performance (*Yao et al., 2021*). For GO Biological Process prediction, network data can help in achieving high prediction accuracy (e.g. biological pathways, genetic interactions, protein interactions). An ongoing challenge remains how to process this type of data contained in large, complex networks in order to combine it with more structured data (e.g. CATH FunFam inferred GO annotations). One method that shows some success is the Multi-Modal Auto Encoder (MMAE), which processes network data to a lower dimensional latent space that can then be integrated with other data types more effectively (*Gligorijevic et al., 2018*).

Here, we combine complementary phenomics data, predicted annotations from integrated machine-learning methods as well as curated and experimental PomBase annotations to provide diverse clues about gene functions and functional associations in fission yeast. We obtained 103,520 quantitative phenotype datapoints for 3492 non-essential genes across 131 diverse conditions. By applying new machine-learning methods, we also report a high-confidence set of 1675 novel GO term associations for 783 genes, derived by combining our phenotype data and functional predictions with PomBase annotations. This rich and wide-ranging functional information for *S. pombe* proteins, including the priority unstudied proteins and many poorly characterized proteins, provides a rich framework and testable hypotheses for follow-on studies.

## Results and discussion
### Overview of the study

We performed colony-based phenotyping of the deletion mutants for the non-essential *S. pombe* genes in response to different nutrient, drug, and stress conditions. This phenomics dataset provides both functional clues and insights into functional relationships from phenotype correlations. In parallel, we developed a meta-predictor (NET-FF) that combines established methods for exploiting protein-network and protein-family data to predict new GO associations for *S. pombe* genes. We then integrated our phenomics data and NET-FF predictions with the gold-standard annotations in PomBase,

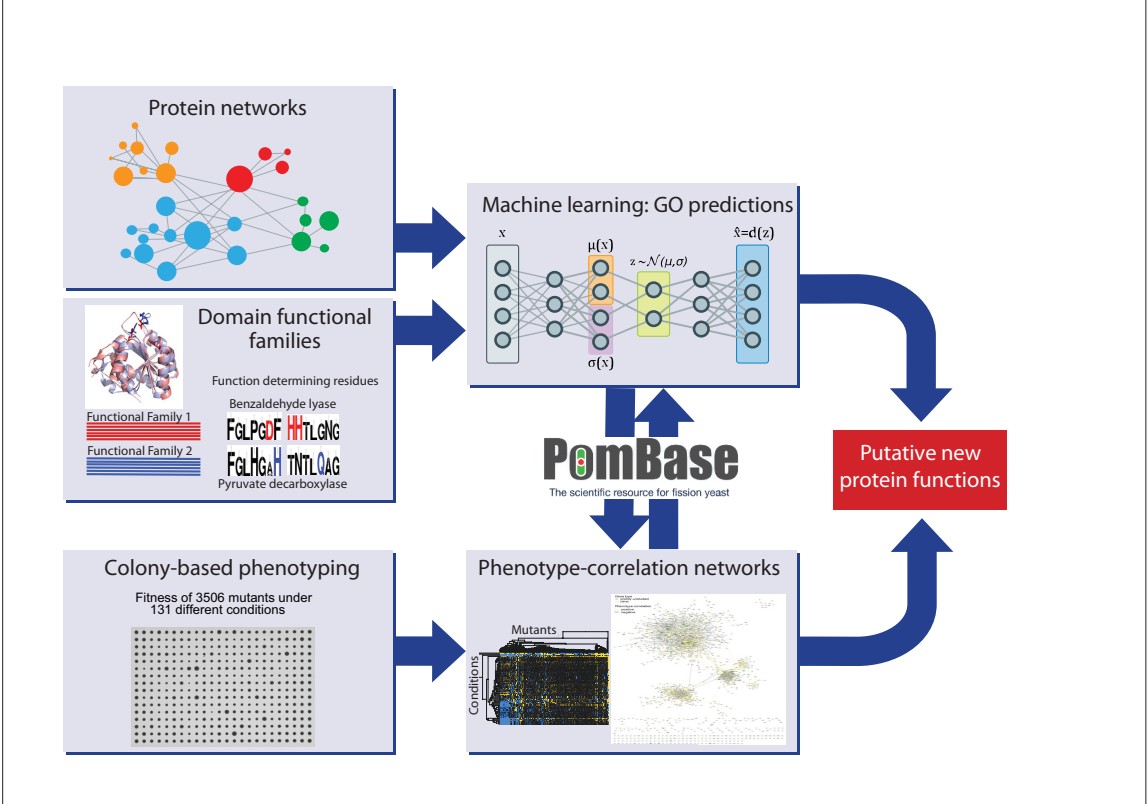

**Figure 1.** Schematic overview of study design. Experimental and computational tasks along with the relationships between the different aspects of the work to generate new information on protein function.

reflecting extensive experimental and curation work (*Harris et al., 2021*; *Lock et al., 2019*), to validate our data and identify novel protein functions and associations. *Figure 1* provides an overview of this study.

## Colony-based phenotyping of deletion mutants in multiple conditions

We combined two *S. pombe* deletion-mutant libraries (*Kim et al., 2010*; *Chen et al., 2015*) and crossed out all auxotrophy mutations, allowing us to screen 3509 non-essential genes in prototroph strains that only contained the query mutants. We measured colony growth as a fitness proxy under different conditions using *pyphe,* our python package for phenomics analyses (*Kamrad et al., 2020*; *Rodriguez-Lopez et al., 2022*). The mutant strains were arrayed in 1536-colony format, including a 384-colony grid of wild-type strains as control. We assayed the deletion mutants in response to 131 diverse environmental conditions, both benign and stressful, including different nutrients and drugs as well as oxidative, osmotic, heavy-metal, protein-homeostasis, and DNA-metabolism stresses. We also assayed some combined conditions which can reveal additional phenotypes through non-additive effects that are not evident from single conditions (*Rallis et al., 2013*). For some drugs and stresses, we assayed both low and high doses, in which wild-type cell growth is normal or inhibited, respectively, to uncover both sensitive or resistant mutants. In total, these assays produced 2,832,384 data points, providing 103,520 phenotypes across 3492 mutants (*Supplementary file 1*). These data exceed all the combined annotated phenotypes available in PomBase, currently encompassing 92,681 gene-phenotype associations of which 59,382 are unique (status March 2023).

## Growth phenotypes in standard conditions

We first assayed fitness differences in standard, benign conditions: growth in rich or minimal medium at 32 °C. We looked for mutants showing a significant difference in colony growth relative to wild-type control cells, applying a threshold of p<0.05 after Benjamini-Hochberg correction and a colony-size difference of ≥5% compared to wild-type. Among the 3509 deletion mutants tested, 732 and 760

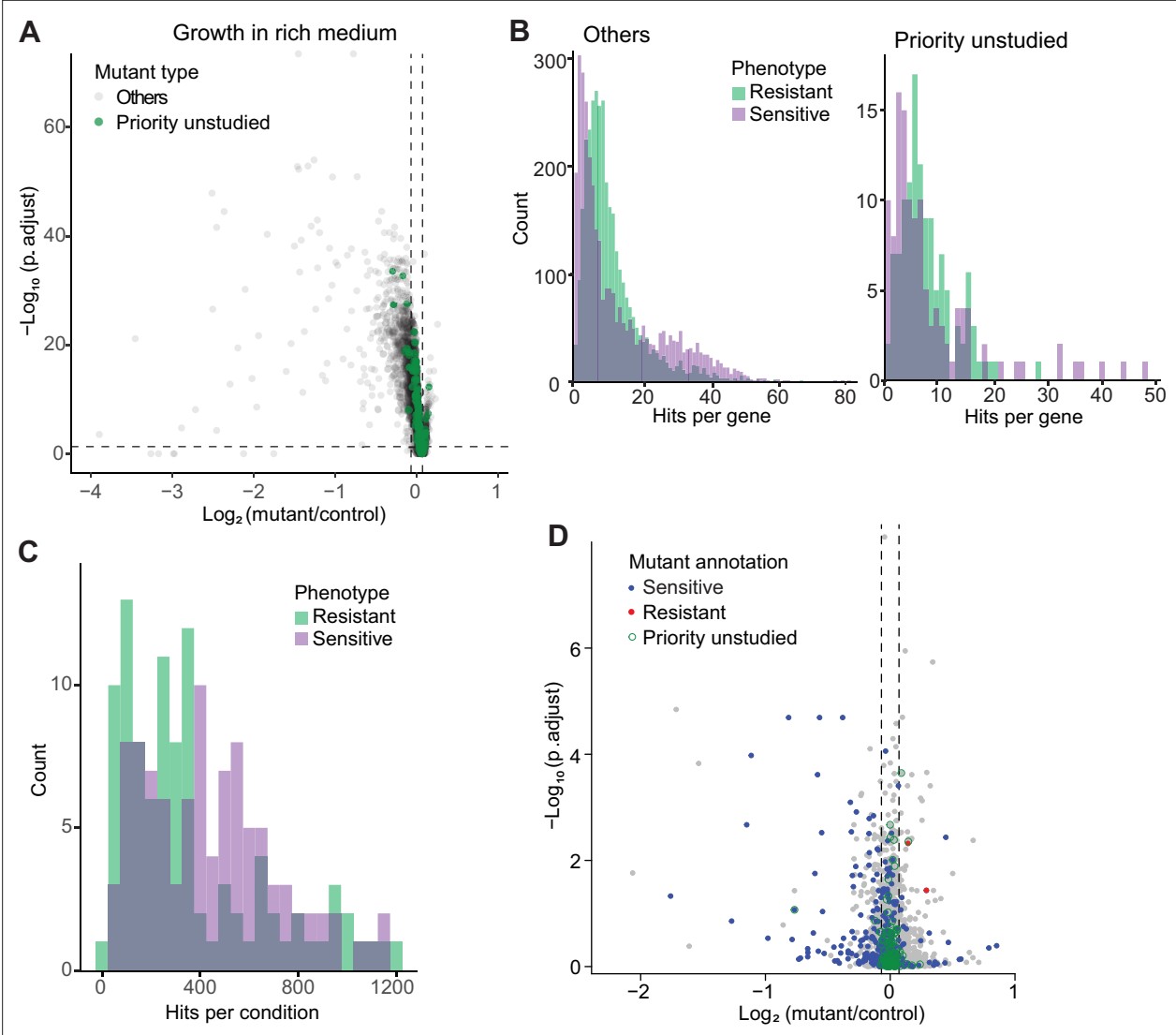

**Figure 2.** Colony-based phenotyping of deletion mutants. (**A**) Volcano plot of mutant colony sizes for priority unstudied genes (green) and all other genes (gray) growing in rich medium. The dashed vertical and horizontal lines show the 5% difference in effect size and significance threshold, respectively. Strains with lower fitness (smaller colonies) are <0 on the x-axis, and those with higher fitness are >0. We applied a significance threshold of 0.05 after Benjamini–Hochberg correction for multiple testing and a difference in fitness of abs(log2(mutant/wild type))>log2(0.05) to call hits based on colony size (n=60). (**B**) Distributions of significant hits per gene for priority unstudied mutants (right) and all other mutants (left) showing faster (green) or slower (purple) colony growth compared to wild-type cells. (**C**) Distributions of significant hits per condition for all mutants showing faster (green) or slower (purple) colony growth compared to wild-type cells. (**D**) Fitness of mutant cells growing in 2% glycerol + 0.01% glucose in this study, plotted as in (**A**). Blue: mutants identified as slow growing in the same condition in our previous screen *Malecki and Bähler, 2016*; red: mutants annotated as 'increased cell population growth on glycerol carbon source' (FYPO:0004167) in PomBase (*Harris et al., 2013*); green rings: mutants in priority unstudied genes. Seventy-nine mutants were slow-growing in both screens (p=1 × 10⁻³⁹; Fisher's exact test, FDR corrected for multiple testing).

The online version of this article includes the following figure supplement(s) for figure 2:

**Figure supplement 1.** Colony-based phenotyping of mutants in benign conditions.

**Figure supplement 2.** Our phenotype association matrix has much higher concordance with STRING than expected by chance.

**Figure supplement 3.** Clusters of mutant phenotype profiles for oxidative-stress conditions.

mutants grew slower than wild-type cells in rich and minimal media, respectively, while 265 and 22 mutants grew faster in the same media (*Figure 2A*; *Figure 2—figure supplement 1A*; *Supplementary file 1*). The two media showed substantial overlap but also distinct responses. As an example, cells deleted for the transcription factor Fil1, which regulates the response to amino acid starvation, showed slower growth in minimal medium, as reported previously (*Duncan et al., 2018*), but faster

growth in rich medium. The differences between the two media were also reflected in distinct functional enrichments (*Figure 2—figure supplement 1B*). For example, mutants that grew slowly only in the minimal medium were enriched in amino-acid biosynthesis and sulfate assimilation, reflecting auxotrophies, while mutants that grew slowly only in the rich medium were enriched in the maintenance and fidelity of DNA replication, possibly reflecting an increased need for quality control in rapidly proliferating cells. Among the mutants analyzed, we paid particular attention to those deleted for so-called 'priority unstudied' genes, which are widely conserved from fission yeast to humans but remain entirely uncharacterized (*Wood et al., 2019*). We could phenotype 124 mutants of the priority unstudied genes, 13 of which grew slower in either or both media while 11 grew faster (*Figure 2A*; *Figure 2—figure supplement 1A*). This finding shows that even under standard conditions some of these mutants readily reveal fitness phenotypes.

We compared our results with the relevant fission yeast phenotype ontology (FYPO) terms annotated in PomBase (*Lock et al., 2019*; *Harris et al., 2013*), including 'increased cell population growth' (253 mutants), and 'decreased vegetative cell population growth,' or 'slow vegetative cell population growth' (1043 mutants). Overall, our screen confirmed 426 (40.8%) of the previously annotated slow-growth mutants but only 23 (9%) of the annotated fast-growth mutants. Besides technical discrepancies, the mutants showing different growth behavior in the different assays may reflect differences in the type of mutant, between solid and liquid media, and/or between mutants growing in isolation or in pools given that a previous growth screen involved parallel mutant profiling by Bar-seq (*Sideri et al., 2015*). Our phenotype data also showed good agreement with our recent study of 238 coding-gene mutants that have been phenotyped along with non-coding RNA mutants (*Rodriguez-Lopez et al., 2022*). Taken together, we identified 940 mutants that grow more slowly and 265 mutants that grow more rapidly in standard media, including 552 and 21 mutants that grow more slowly or rapidly, respectively, in both media.

## Phenotypes in diverse nutrient, drug, and stress conditions

We then assayed fitness differences of the deletion mutants in the presence of various stresses or other treatments, relative to the same mutants growing in standard conditions and normalized for wild-type growth. Because only four repeats were measured per condition, fewer than for the benign conditions, we did not use p-values but applied a more stringent threshold to identify sensitive (slowly growing) or resistant (rapidly growing) mutants, using an effect-size difference of ≥10% compared to the standard condition in the same medium. Overall, we could measure colony growth for 3506 mutants, 3492 (99.6%) of which showed phenotypes in at least one condition, including 55,577 sensitive and 47,943 resistant phenotypes (*Figure 2B*; *Supplementary file 1*). Thirty-eight mutants displayed many different phenotypes in 100–118 conditions (*Supplementary file 1*), pointing to pleiotropic genes that exert multiple functions. All 124 mutants of the priority unstudied genes produced phenotypes in at least one condition, with an average of 19.4 phenotypes per gene compared to an average of 29.9 phenotypes per gene for all other mutants (*Figure 2B*). Moreover, all 129 conditions tested caused phenotypes in 118–2761 mutants (*Figure 2C*; *Supplementary file 1*). Thus, we obtained rich and diverse phenotype data for nearly all mutants tested, including those of priority unstudied genes.

To validate our approach, we compared one condition (growth in medium with glycerol) with previous results from the same condition (*Malecki and Bähler, 2016*). This analysis revealed a strong overall agreement between the two datasets (*Figure 2D*). Differences may reflect the intrinsic variability of high-throughput screens and the more refined normalization procedure applied here (*Kamrad et al., 2020*; *Rodriguez-Lopez et al., 2022*), using wild-type cells as a common reference. We also compared correlations across phenotypes between deleted genes in our phenotype data with protein-protein interaction data in STRING (*Szklarczyk et al., 2021*). This analysis revealed 425 protein pairs that were supported by phenotype correlations, many more than expected by chance (*Figure 2—figure supplement 2*). Thus, phenotype correlations between pairs of knockout strains recapitulate known protein interactions assayed independently, demonstrating the validity of this approach.

## Mutants showing altered resistance to oxidative stresses

These extensive phenotype data provide rich functional information regarding specific genes and processes that negatively or positively affect cellular fitness in a wide range of environmental or

physiological contexts. As an example, we highlight oxidative stress which triggers a widely studied cellular response (*Vivancos et al., 2006*; *Ikner and Shiozaki, 2005*). Our data included mutant fitness under different doses or exposure times of three oxidants: hydrogen peroxide ($H_2O_2$), *t*-butylhydroperoxide (TBH), and diamide. Over 90% of the mutants were sensitive or resistant under at least one of these conditions (*Figure 2—figure supplement 3*). Sensitivity or resistance was often specific to the oxidant dose. For example, of the 495 and 220 sensitive mutants in the low or high dose of $H_2O_2$, respectively, only 96 mutants were sensitive in both doses. Even more pronounced, of the 304 and 95 resistant mutants in the low or high dose of $H_2O_2$, respectively, only 25 mutants were resistant in both doses. This finding is consistent with studies showing that different stress-response pathways and gene-expression programs are launched in different doses of $H_2O_2$ (*Vivancos et al., 2006*; *Chen et al., 2008*).

We generated conservative lists of mutants that were sensitive or resistant in at least 3 of the 6 oxidative-stress conditions tested, resulting in 610 sensitive and 365 resistant mutants (*Supplementary file 1*). Using AnGeLi (*Bitton et al., 2015*), we looked for functional enrichments among these lists with respect FYPO (*Harris et al., 2013*) and GO terms (*Carbon et al., 2021*). The 610 genes leading to oxidative-stress sensitivity, when deleted, showed diverse enrichments, including 434 genes ($p=8.2 \times 10^{-7}$) involved in metabolic processes, particularly in the negative regulation of metabolism, 30 genes ($p=5.0 \times 10^{-7}$) involved in endosomal transport, 79 genes ($p=1.1 \times 10^{-11}$) that regulate transcription, 503 genes ($p=8.4 \times 10^{-50}$) required for normal growth, 104 genes ($p=2.0 \times 10^{-20}$) with cytoskeletal functions, and all six genes ($p=8.9 \times 10^{-5}$) of the adenylate cyclase-activating G-protein coupled receptor signalling pathway. Surprisingly, the 610 genes were not significantly enriched for genes induced during oxidative stress, but they included 42 genes ($p=3.3 \times 10^{-5}$) encoding ribosomal proteins that are down-regulated during oxidative stress (*Chen et al., 2008*). This finding is consistent with studies from budding yeast reporting a poor correlation between the genes induced in response to stress and the genes required to survive that stress (*Giaever et al., 2002*; *Berry and Gasch, 2008*). It also raises the interesting possibility that certain ribosomal proteins have specialized roles in stress protection. The 610 sensitive mutants included only 64 genes ($p=1.3 \times 10^{-16}$) already annotated with the FYPO term 'sensitive to hydrogen peroxide.' However, 445 of the 610 genes ($p=3.1 \times 10^{-57}$) are annotated as 'increased sensitivity to chemical,' indicating that oxidant-sensitive mutants also tend to be sensitive to other compounds. In contrast to the sensitive mutants, the 265 resistant mutants showed no informative functional enrichments, and only eight of these genes have been previously annotated to the FYPO term 'resistant to hydrogen peroxide.' Thus, the present study uncovered over 900 proteins not previously linked to oxidative stress in *S. pombe*, including 18 priority unstudied proteins.

## Clustering of mutants with similar phenotype profiles

To explore functional signatures of the deletion mutants across all conditions, we applied *k*-medoids clustering of our phenotype data, revealing eight main clusters (*Figure 3A*). Analyses with Metascape (*Zhou et al., 2019b*) and AnGeLi (*Bitton et al., 2015*) showed that each of these eight clusters features distinct functional enrichments, respectively, in different GO and KEGG pathway terms (*Figure 3B*; *Supplementary file 1*) and/or in genes from published expression or phenotyping studies. Examples of significant functional enrichments for these clusters are described below. Cluster 1 (293 genes, 12 priority unstudied) is enriched for genes showing high expression variability across different conditions (*Pancaldi et al., 2010*) and for genes induced during meiotic differentiation (*Mata et al., 2002*) and in response to TORC1 inhibitors (*Rodríguez López et al., 2020*). Cluster 2 (570 genes, 20 priority unstudied) is enriched for phenotypes related to cell mating and sporulation, e.g., 'incomplete cell-wall disassembly at cell fusion site' or 'abnormal shmoo morphology' (*Dudin et al., 2017*). Cluster 3 (806 genes, 29 priority unstudied) is enriched in endosome-vacuole, peroxisome, and protein catabolism functions and in core environmental stress response genes (*Chen et al., 2003*). Cluster 4 (454 genes, 19 priority unstudied) is enriched in functions related to mitochondrion organization and oxidoreductase activity as well as in genes affecting the chronological lifespan (*Romila et al., 2021*). Cluster 5 (633 genes, 24 priority unstudied) is enriched for citrate-cycle and sulphur-relay pathways as well as phenotypes related to abnormal cell growth and increased sensitivity to chemicals, based on FYPO terms from several studies (*Harris et al., 2013*). Cluster 6 (210 genes, four priority unstudied) is enriched in many different functions related to actin and microtubule cytoskeletons, cell

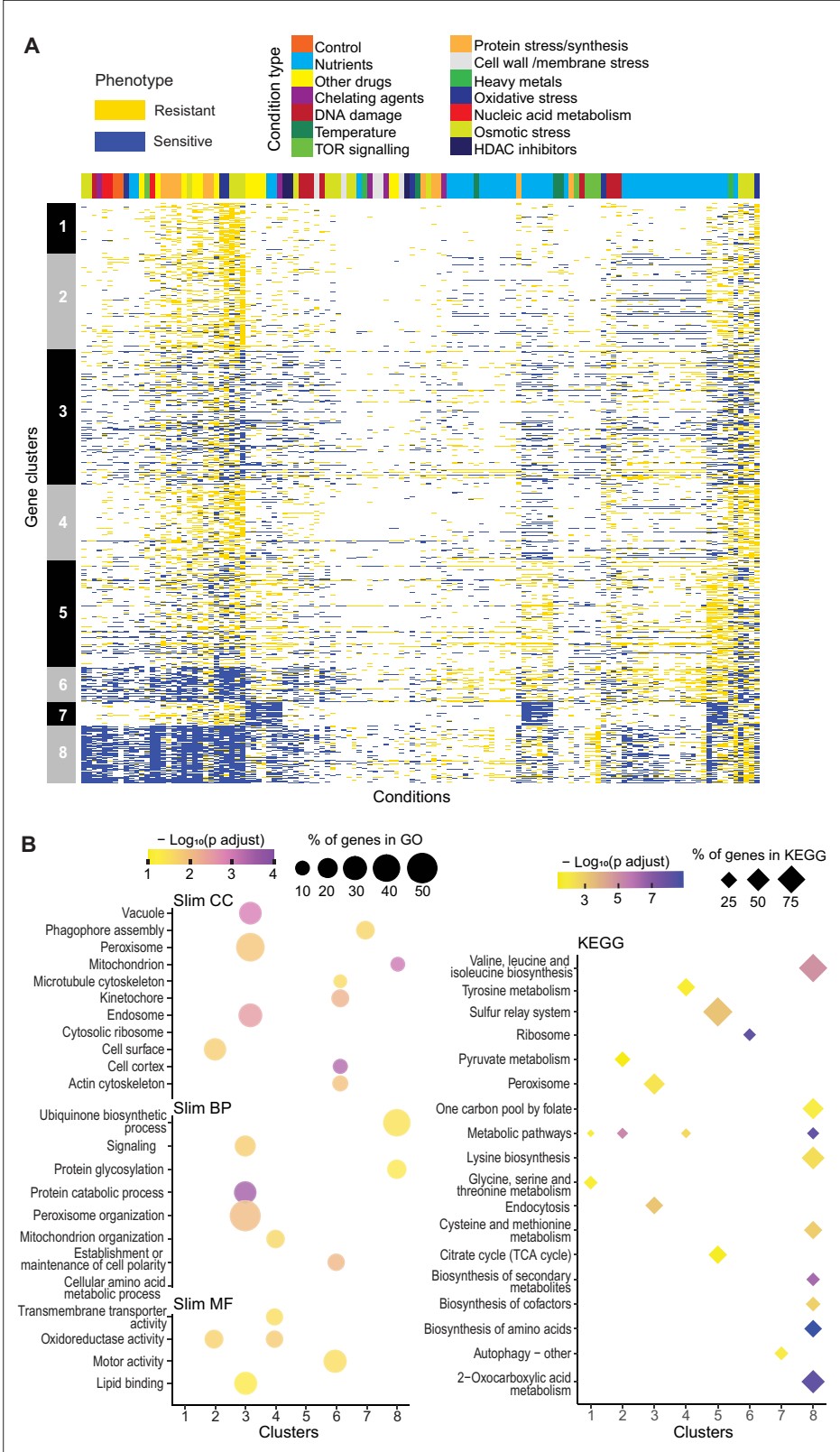

**Figure 3.** Mutant phenotype profiles. (**A**) Clustering of discretized data for 3449 deletion mutants (rows) in 98 conditions (columns), following recommendations from the R package microbial Phenotypes. For discretizing the phenotype data, we classed mutants as either resistant (yellow), sensitive (blue), or similar (white) relative to their fitness in the corresponding control condition, applying an effect-size threshold of ≥10% and, for the two benign

*Figure 3 continued on next page*

*Figure 3 continued*

conditions, also a significance threshold of p<0.05 (*Supplementary file 1*). To improve the clustering, we removed conditions and mutants that produced <4 phenotypes, the three conditions that produced >2400 phenotypes, and genes with >50% of missing values. Different types of conditions are color-coded on top. Indicated at left are the eight main clusters, calculated using *k*-medoids in R (pam function) and a distance of 1-Pearson correlation. Using the function fviz_nbclust (R package factoextra), we determined that 8 is an optimum number of clusters. (**B**) Functional enrichments of the eight gene clusters in (**A**) using GO slim categories (*Carbon et al., 2021*) for Cellular Component (CC), Biological Process (BP), and Molecular Function (MF) (left, using SystempipeR) as well as KEGG pathways (right, using g:Profiler) (*Raudvere et al., 2019*).

morphogenesis, chromatin segregation, and ribosomes as well as in ribosomal-protein gene clusters that are repressed in oxidative stresses (*Chen et al., 2008*). Consistent with these GO enrichments, Cluster 6 is also enriched in phenotypes related to abnormal cell shape and cytoskeleton organization, based on FYPO terms from numerous studies (*Harris et al., 2013*). Cluster 7 (137 genes, seven priority unstudied) is enriched in autophagy-related functions. Cluster 8 (346 genes, five priority unstudied) is enriched in metabolism-related functions like amino-acid biosynthesis and mitochondrion, in genes with slow-growth (*Sideri et al., 2015*) and caffeine-sensitive (*Rallis et al., 2014*) phenotypes as well as in genes showing high levels and stability of mRNAs (*Lackner et al., 2007*; *Hasan et al., 2014*). These enrichments allow inferring possible functions for the mutants in unknown genes that show similar phenotypic profiles to genes functioning in known processes through the principle of 'guilt by association.'

## The NET-FF meta-predictor combines protein-network and -family data to associate GO terms with *S. pombe* proteins

We developed an integrated pipeline (NET-FF) to predict GO terms for *S. pombe* proteins by combining two independent established approaches, NetHom and CATHPredictGO. NetHom uses a machine-learning method to combine network features derived using DeepNF (*Gligorijevic et al., 2018*) with homology features from CATH protein functional families (FunFams) (*Sillitoe et al., 2021*). CATHPredictGO also uses homology features but exploits this data with an independent, complementary approach. To obtain our final set of predictions, NET-FF combines the predicted protein functions from the NetHom and CATHPredictGO predictors to increase confidence and coverage of the predictions (see Methods for details of NetHom, CATHPredictGO and their integration in NET-FF).

Using NET-FF, we predicted a total of 2,390,915 GO terms for *S. pombe* proteins. Filtering for high-scoring predictions (>0.7 from a maximum of 1.0) and applying a taxon constraint resulted in 56,594 GO terms (*Figure 4—figure supplement 1*). These terms comprised 42,808 Biological Process (BP) and 13,786 Molecular Function (MF) terms, across 2852 *S. pombe* proteins, including 53 priority unstudied proteins (*Supplementary file 2*). Below, we discuss the predictions from NET-FF and their validation. Subsequently, we combine our NET-FF predictions and phenotype-correlation data with curated annotations in PomBase to derive novel functional associations of *S. pombe* proteins.

## Systematic comparison of NET-FF predictions and PomBase annotations

To assess the validity of our 56,594 high-scoring NET-FF predictions, we determined the similarity between the highly curated GO terms annotated by PomBase and those predicted by NET-FF. Although only 60% of the PomBase annotations are supported by direct experimental evidence, a further 15,438 (34.5%) are curated from inferred annotations, providing a rich knowledgebase for the *S. pombe* community. To compare the PomBase annotations with NET-FF predictions, we modified a standard approach for comparing two GO terms (GOGO method; 79) to perform a pairwise comparison using the same gene with different GO terms sets, e.g., GOGO(gene A$_{PomBase}$ - gene A$_{NET-FF}$). This approach provided a GO semantic similarity between the two datasets for a given gene, with 0 having completely different GO terms and 1 having identical GO terms. *Figure 4A* shows the distribution of GOGO scores obtained by comparing the two datasets and a randomized control obtained by shuffling GO terms between genes. Although the datasets differ (median difference GOGO-BP=0.52, median difference GOGO-MF=0.64), the real data comparison gives a higher GOGO semantic similarity when compared to random. *Supplementary file 2* provides the list of predictions from NET-FF above our cut-off, including those for which our predictions agree or are very similar to those

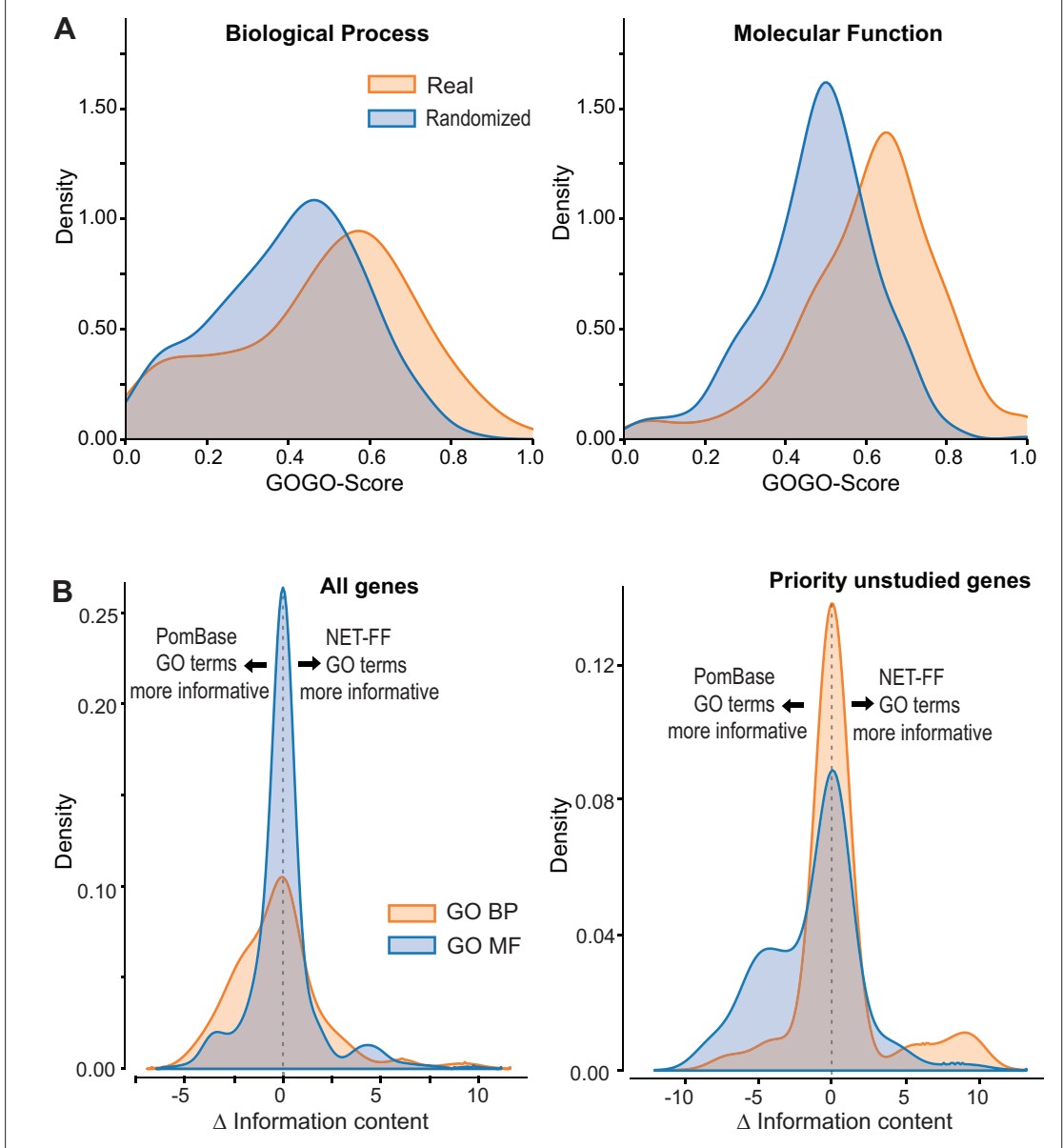

**Figure 4.** NET-FF predictions of gene ontology (GO) terms for *S. pombe* proteins. (**A**) GOGO semantic similarity scores for GO terms predicted by NET-FF and GO terms annotated by PomBase, both for real (orange) and randomized (blue) datasets from NET-FF. Left: Biological Process data, right: Molecular function data. (**B**) Information content distribution for GO terms predicted by NET-FF and GO terms annotated by PomBase, both for GO Biological Process (orange) and GO Molecular Function (blue) terms. Left: dataset for all genes, right: subset for priority unstudied genes.

The online version of this article includes the following figure supplement(s) for figure 4:

**Figure supplement 1.** Filtering of predictions by information content and probability value.

**Figure supplement 2.** Performance of NET-FF predictor by source data.

annotated independently in PomBase. We conclude that NET-FF predictions are more similar to the available annotations in PomBase than random.

Aside from false predictions in NET-FF, the difference in GO terms predicted by the GOGO analyses may suggest the presence of more specific GO terms in either dataset, i.e., more informative annotations that occur deeper in the GO hierarchy. We calculated the Information Content (IC) for all GO terms in PomBase (Methods) and, for a given gene, extracted the GO with the highest IC. We then calculated for each gene the ΔIC as ($IC_{NET-FF}$ - $IC_{Pombase}$), with positive or negative results indicating that NET-FF or PomBase, respectively, have the more informative GO for this gene. *Figure 4B* shows the

distribution of ΔIC for the BP and MF gene datasets that are in common between the NET-FF predictions and PomBase annotations. For MF, 30% of the PomBase annotations were more informative than NET-FF predictions, while 28% of the NET-FF predictions were more informative than PomBase annotations (*Figure 4B*). For BP, the proportion of more informative PomBase annotations (51%) was higher than for more informative NET-FF predictions (33%). Notably, the situation was different for the priority unstudied genes, where PomBase was more informative with respect to MF (42.5% vs 13.5%), while NET-FF was more informative with respect to BP (14.9% vs 8.5%) (*Figure 4B*). Given that BP terms are key to understand the cellular roles of proteins, this result indicates that the NET-FF predictions provide valuable new clues for the function of priority unstudied proteins.

PomBase annotations are subject to manual curation including extensive literature searches for available gene information (*Harris et al., 2021*; *Lock et al., 2019*). Therefore, we naturally expected the proportion of informative PomBase annotations to be higher than for NET-FF predictions. Of the 56,594 NET-FF predictions, 22,060 had a high information content (IC >5), reflecting more specific GO terms (*Figure 4—figure supplement 1*; *Supplementary file 2*). Of these, 9374 terms were identical to PomBase GO terms with experimental annotations for the same genes. This overlap in predictions is not that surprising, because NET-FF was trained on GO terms with experimental annotations but not on terms inferred from electronic annotations. Notably, of the remaining 12,686 NET-FF predictions, 6052 (48%) were identical to curated or electronically inferred PomBase annotations for the same genes (PomBase GO version 13/01/23). This substantial overlap provides compelling validation of NET-FF.

We further filtered the 6634 NET-FF predictions that were not identical to PomBase annotations to remove 276 GO terms with more informative (deeper) annotations in PomBase, 498 GO terms that became obsolete, and 4185 GO terms that were up-propagated and thus redundant with more informative terms. This resulted in 1675 novel NET-FF GO predictions for 783 *S. pombe* genes, including 47 predictions for 23 priority unstudied genes (*Supplementary file 2*). Of these novel predictions, 1481 terms are in parts of the GO tree not previously assigned to these genes and 194 terms provide more informative annotations than PomBase. Some of the novel predictions contain 'regulation of,' suggesting a role in a regulatory process that could be direct or indirect. The substantial overlap between the NET-FF predictions and the high-confidence annotations available in PomBase supports the notion that the 1675 novel predictions can provide unique functional clues for the associated genes.

## Manual comparison of selected NET-FF predictions with GO terms and phenotypes annotated in PomBase

We also manually validated a selection of the NET-FF GO terms using the rich extant information in PomBase, including both GO and FYPO terms (*Harris et al., 2021*; *Lock et al., 2019*; *Harris et al., 2013*). We selected 312 genes with predicted GO terms that relate to broad cellular processes: aging/ cell death, growth/response to nutrients, reproduction, recombination, cold/heat response, histone acetylation/deacetylation, DNA damage/repair, autophagy, and actin (*Supplementary file 2*). We checked to what extent aspects of our NET-FF predictions for these genes had already been annotated by searching PomBase for GO and/or FYPO terms related to the predicted GO terms. For example, 'decreased cell population growth at low temperature' (FYPO:0000080) relates to 'cold acclimation' (GO:0009631) and 'response to cold' (GO:0009409). This analysis identified 165 of the 312 genes (52.9%) with NET-FF GO terms similar to the PomBase FYPO and/or GO annotations (*Supplementary file 2*). Moreover, several NET-FF GO terms were supported by our colony-based phenotype data. For example, mutants in 4 of 13 genes with predictions related to cold/heat response showed phenotypes at low and/or high growth temperatures, mutants in 3 of 6 genes with predictions related to histone acetylation/deacetylation showed phenotypes in conditions affecting those processes, mutants in 31 of 34 genes with predictions related to growth/nutrient response showed altered growth in different nutrients, and mutants in 30 of 31 genes with predictions related to DNA damage/repair showed phenotypes in conditions causing different types of DNA damage. Thus, this independent experimental evidence further supports our NET-FF predictions.

For the remaining 147 of the 312 selected genes (47.1%), the NET-FF GO terms represent novel predictions, revealing putative functions that were not previously known (*Supplementary file 2*). The proportions of novel predictions varied greatly for different process-related categories. As expected,

processes that are well-studied by *S. pombe* researchers were already well-captured in PomBase, with relatively few novel predictions. For example, only 2 of 13 genes (15.4%) in the category 'Histone acetylation/deacetylation' and 7 of 42 genes (16.7%) in the category 'DNA damage/repair' were not annotated as such in PomBase. In contrast, 11 of 12 genes (91.7%) in the category 'aging/cell death' and 26 of 44 genes (59.1%) in the category 'growth/response to nutrients' were not annotated as such in PomBase. Five of the genes associated with novel GO-term predictions were priority unstudied. The new predictions allow us to make specific functional inferences about these unknown proteins (*Supplementary file 2*). For example, *SPAC15A10.10* was predicted to function in the regulation of nuclear division and actin-related processes, and *SPAC25B8.08* in actin-dependent ATPase activity and organelle localization. Notably, the deletion of *SPBC2D10.03c*, predicted to function in the response to heat, leads to decreased growth at high temperatures (*A Rahaman et al., 2018*), thus validating the prediction. Overall, the NET-FF predictions recapitulate GO and/or phenotype predictions in PomBase for half of the genes in the selected broad categories, thus adding confidence to our method.

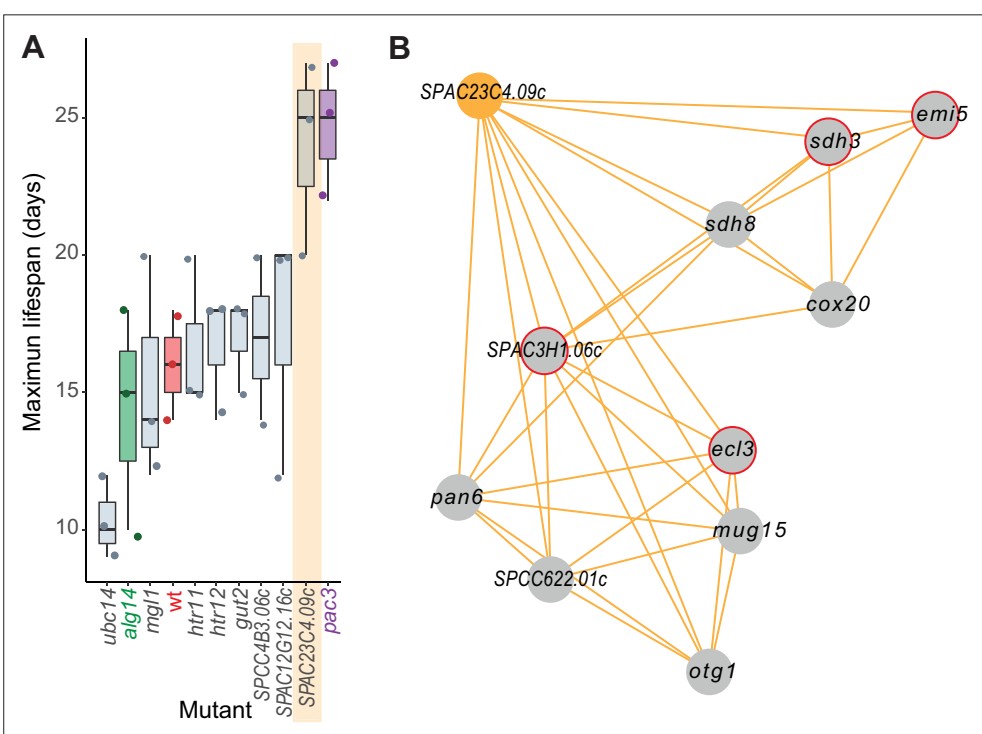

**Figure 5.** Experimental validation of new aging-related gene reflecting its NET-FF gene ontology (GO) predictions and phenotype-correlation network. (**A**) Maximal chronological lifespan (CLS) for eight selected deletion mutants of genes with NET-FF GO terms related to aging/cell death and DNA damage repair, measured using a high-throughput CLS assay (*Romila et al., 2021*). Known short- and long-lived mutants are highlighted in green and purple, respectively. Of the eight mutants, *ubc14* and *htr12* were known to be short- and long-lived, respectively (*Romila et al., 2021*). As controls, the maximal CLS of known short-lived (*alg14*) and long-lived (*pac3*) mutants (*Romila et al., 2021*) and wild-type cells (wt, red) are included. Three independent biological repeats were performed with actual data points shown as dots. Orange band: mutant in priority unstudied gene (*SPAC23C4.09c*) showing strong longevity phenotype. Data available in *Figure 5—source data 1*. (**B**) Cluster from Cytoscape network (see *Figure 6*) representing phenotype correlations between the *SPAC23C4.09c* gene-deletion mutant and ten other mutants. Orange edges show positive phenotype correlations, and red circles indicate genes with previously known CLS phenotypes (*Romila et al., 2021*; *Ohtsuka et al., 2011*). Details are discussed in the main text.

The online version of this article includes the following source data for figure 5:

**Source data 1.** Data for chronological lifespan (CLS) plots in *Figure 5A*.

## Experimental validation of genes predicted to function in cellular aging

The general category 'aging/cell death' included the largest proportion of genes linked to NET-FF predictions not available in PomBase (11 of 12 genes; 91.7%). This result raises the possibility that these are novel genes affecting the chronological lifespan (CLS) of stationary phase *S. pombe* cells, a model for cellular aging. We determined the maximal CLS for eight deletion mutants of these genes and control strains (*Figure 5A*). Seven of these mutants showed subtle to strong effects on the CLS, including one mutant in a priority unstudied gene (*SPAC23C4.09c*) featuring a strong longevity phenotype. In contrast to this result obtained using stationary-phase cells, the *SPAC23C4.09c* mutant has been reported to reduce the CLS in quiescent cells (*Zahedi et al., 2020*). We have shown before that some genes exert opposite effects on the CLS in stationary-phase cells, limited by glucose, and in quiescent cells, limited by nitrogen (*Sideri et al., 2015*). Besides cell death, *SPAC23C4.09c* was predicted to function in DNA-damage repair (*Supplementary file 2*), a process important for aging (*Schumacher et al., 2021*). Notably, the *SPAC23C4.09c* gene formed a tight phenotype-correlation network with ten other genes, including four with known CLS phenotypes (*Figure 5B*). Several of these genes are involved in respiration and carbon metabolism, including *otg1* (galactosyltransferase), *cox20* (cytochrome c oxidase assembly protein), *sdh8* (mitochondrial respiratory chain complex II assembly factor), *sdh3* (succinate dehydrogenase cytochrome b subunit), and *emi5* (succinate dehydrogenase complex assembly). Consistent with these interactions, the *SPAC23C4.09c* mutant grows slowly on non-fermentable media (*Malecki et al., 2016*). Together, these experimental results validate the computational predictions by confirming new genes with roles in cellular ageing and illustrate how our predictions and the phenomics data complement each other to reveal new gene functions.

## Combining NET-FF predictions with phenotype correlation data to validate functional associations and derive higher-confidence predictions

To further filter our set of 22,060 NET-FF predictions with high scores and high information content, we identified pairs of genes showing high similarity both in their GO-term predictions (GOGO scores) and in their deletion-mutant phenotypes (PHEPHE scores). This approach should provide a higher confidence subset of gene pairs for which the two independent, orthogonal methods agree. For each gene pair with a predicted GOGO BP score >0.5, the GOGO score was multiplied by the PHEPHE score of the same gene pair (obtained from Pearson correlation of phenotype profiles across conditions). If the combined product of GOGO and PHEPHE scores was >0.5, we added the gene pair link to a network. This analysis resulted in a much larger network for the gene pairs with high GOGO and PHEPHE scores than for random gene pairs (*Figure 6—figure supplement 1A*). *Supplementary file 2* provides a list of the gene pairs with high GOGO and high PHEPHE scores, both for BP terms (553 gene pairs) and MF terms (606 gene pairs). The agreement between these two independent approaches increases the confidence in both our NET-FF predictions and phenotype data.

For comparison, we used an analogous approach to identify sets of gene pairs that showed high GO similarity in their PomBase annotations. *Supplementary file 2* provides a list of these gene pairs with high GOGO and high PHEPHE scores, both for BP terms (12,587 gene pairs) and MF terms (20,433 gene pairs). As expected, these are larger gene sets than for the NET-FF predictions, generating a much larger network for these gene pairs than random pairs (*Figure 6—figure supplement 1B*). This analysis adds further confidence to our phenotype data.

## Integrated analysis of phenotype-correlation networks and GOGO similarities

We used our phenotype-correlation data to construct a Cytoscape network consisting of 138 clusters (*Supplementary file 2*). Given that the phenotype-correlation clusters are enriched in genes with similar functions, we looked for gene pairs with similar GO terms (GOGO score >0.7) based on NET-FF predictions and/or PomBase annotations (*Figure 6A–C*; hatched blue and purple edges, respectively). As expected, the clusters contained many more high GOGO gene pairs based on PomBase annotations, but the NET-FF predictions added some new information (*Supplementary file 2*). An example is the unique NET-FF-based link between *rpl1001* and *hse1* in Cluster 31 (*Figure 6B*). In many cases, high GO similarities linked gene pairs that were not directly linked via high phenotype correlations although they were part of the same cluster, thus independently supporting their cluster membership.

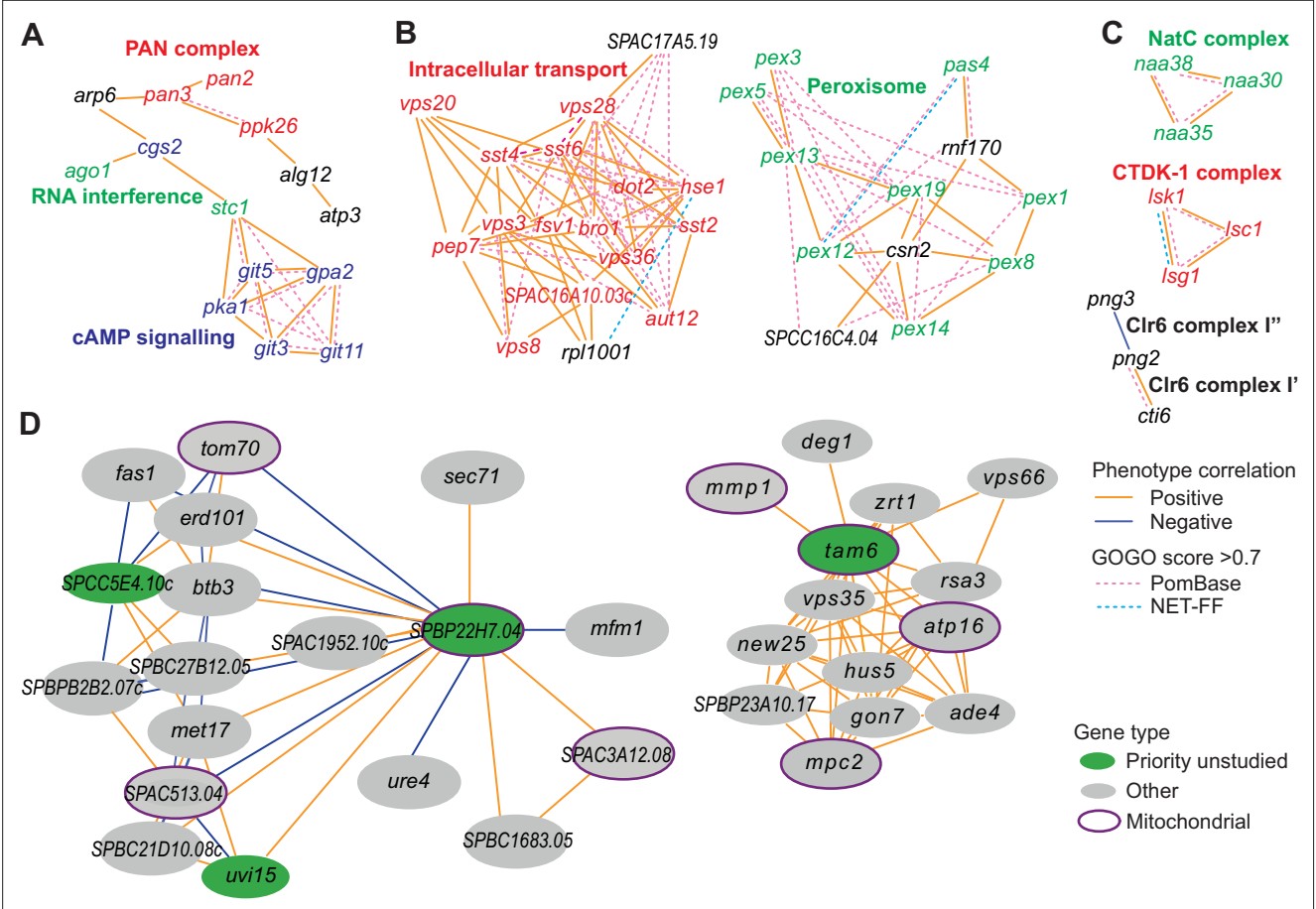

**Figure 6.** Example phenotype-correlation clusters from Cytoscape network. Positive and negative phenotype correlations across the 131 conditions are indicated as orange and blue edges, respectively. Gene pairs with high gene ontology (GO) similarities based on PomBase annotations or NET-FF predictions are indicated as hatched pink and bright blue edges, respectively. (**A**) Cluster 13 consists of a branch with all members of the PAN complex (red) and a branch containing six genes involved in cAMP signaling (blue), along with two genes involved in RNA interference (green). (**B**) Cluster 31 enriched for genes involved in vacuolar and endosomal transport (red) and Cluster 22 enriched for genes with peroxisome functions (green). (**C**) Three clusters linking members of the same protein complexes, involved in protein maturation (NatC, green), regulation of transcription elongation (CTDK-1, red), and histone deacetylation (Clr6, black). The data reflect different Clr6 sub-complexes, with *png2* and *cti6* being positively correlated (Clr6 complex I') but *png2* and *png3* being negatively correlated (Clr6 complex I") (*Zilio et al., 2014*). (**D**) Two networks containing priority unstudied genes (green), including those with implicated mitochondrial functions (green with purple ellipses), together with their nearest neighbors based on phenotype correlation, which includes genes with known mitochondrial functions (gray with purple ellipses). For simplicity, high GOGO pairs are not indicated in these networks.

The online version of this article includes the following figure supplement(s) for figure 6:

**Figure supplement 1.** Gene ontology (GO) association networks show higher concordance with our phenotype data than expected by chance.

**Figure supplement 2.** Enrichment of high GOGO gene pair edges in nine clusters from the Cytoscape clustering of phenotype correlations, containing at least 15 genes.

**Figure supplement 3.** Phenotype-correlation clusters from the entire Cytoscape network.

Of the nine clusters containing at least 15 genes, six showed more high GOGO gene pairs than expected by chance based on NET-FF predictions (*Figure 6—figure supplement 2*). *Supplementary file 2* contains information on the clusters whose high GOGO gene pairs are enriched for specific GO terms based on NET-FF, PomBase, and/or the combination of NET-FF and PomBase.

We then examined whether the inclusion of the NET-FF predictions led to additional GO term enrichments of the clustered data over existing PomBase annotations. Based on PomBase, 308 GO terms were significantly enriched, while the NET-FF predictions added a further 63 GO-term enrichments, including 32 GO terms directly based on NET-FF and 31 GO terms becoming only enriched when combining PomBase and NET-FF terms (*Supplementary file 2*). Together, these analyses show

that the NET-FF predictions can add new information to the phenotype clusters, and the concordance between high GOGO gene pairs and phenotype correlations increases confidence in both the NET-FF predictions and phenotype-correlation data.

The phenotype-correlation network allows for a fine-grained, nuanced analysis of functional relationships (*Rodriguez-Lopez et al., 2022*). Based on the genes with known roles, the clusters were functionally coherent based on gene enrichment analyses and manual inspection, reflecting protein complexes and/or biological pathways. For example, Cluster 13 consisted of a branch with all members of the PAN complex, a polyA-specific ribonuclease, and another branch containing six genes involved in cAMP signaling, along with two genes that function in RNA interference (*Figure 6A*). Consistent with the latter branch, published reports suggest a link between cAMP signaling and RNA interference (*Piazzon et al., 2012*; *Liu et al., 2021*; *Mortensen et al., 2011*). Cluster 31 and Cluster 22 contained mostly genes involved in vacuolar/endosomal transport and peroxisome function, respectively, along with poorly characterized genes (*Figure 6B*). Some clusters linked members of protein complexes and even reflected different sub-complexes (*Figure 6C*). Of the 124 priority unstudied genes, 70 showed phenotype correlations with at least one characterized gene (*Supplementary file 2*). For example, five priority unstudied genes implicated in mitochondrial functions correlated with genes that play known roles in mitochondria. *Figure 6D* shows two such networks including priority unstudied genes along with their nearest neighbors, placing the unstudied genes in a functional context based on phenotype correlations. *Figure 6—figure supplement 3* provides the entire Cytoscape network which can be interactively explored as specified in the legend. Thus, these phenotype-correlation clusters can reveal unexpected relationships between different cellular processes and provide functional clues for poorly characterized or unstudied proteins.

## Conclusions

We applied a phenomics approach using colony-based assays for 3509 deletion mutants in all non-essential genes of fission yeast under many different conditions, including different nutrients and drugs as well as oxidative, osmotic, heavy-metal, protein-homeostasis and DNA-metabolism stresses. All 131 conditions tested led to growth phenotypes in over 100 mutants. We detected phenotypes in at least one condition for 99.6% of the mutants, including all 124 mutants of priority unstudied genes for which we obtained data. Together, these rich data exceed all the phenotypes currently annotated in PomBase. By combining two orthologous gene-function prediction methods (NET-FF), we predicted 22,060 GO terms with high scores and high information content across 2167 genes. Compared to existing PomBase annotations, these GO predictions are more similar than random and of comparable information content. Moreover, our phenotype-correlation data and NET-FF predictions showed a strong overall concordance with the PomBase GO annotations and provide new functional clues for many genes. The agreement between the independent phenomics, curation, and computational approaches increases the confidence in both our NET-FF predictions and phenotype data. Notably, 15,426 NET-FF predictions are identical to PomBase GO annotations for the same genes, while 1675 predictions provide novel GO associations for 783 genes, including 47 predictions for 23 priority unstudied proteins. Experimental validation based on selected predictions revealed new proteins involved in cellular aging and showed that combining the NET-FF predictions and phenomics data can reveal new protein functions. Integrated analyses revealed good agreement between phenotype-correlations and GO similarities for gene pairs, with the NET-FF predictions adding some unique information to the PomBase annotations. These wet and dry approaches provide a rich framework to better understand the functional relationships between proteins and to mechanistically dissect the roles of proteins in physiologically relevant conditions.

## Methods
### Yeast strains

Yeast strains used in this study are described in *Supplementary file 1*; most of the strains are derived from a prototroph version of the Bioneer V.5 deletion collection (*Kim et al., 2010*) generated as described (*Malecki and Bähler, 2016*), and the rest of the strains were kindly provided by Kathy Gould (*Chen et al., 2015*). Prior to phenotyping, the strains were arranged into a 1536 format with a grid of a control wild-type strain (*h- 972*) as described before (*Kamrad et al., 2020*). Cells were grown on

standard Edinburgh minimal medium (EMM) or rich yeast extract medium with supplements (YES) as specified.

## Colony-based phenotyping of deletion-mutant library

The conditions used for phenotyping are provided in *Supplementary file 1*. For phenotypic screens, plates containing 1536 colonies were pinned onto YES plates using a RoToR arraying robot (Singer Instruments) and incubated for 24 hr at 32 °C; these plates served as templates. Strains from the template plates were pinned onto the phenotyping plates using 1536-pin pads and applying 4% pressure to minimize the biomass inoculated onto each plate. Each template plate was used to inoculate 4–6 phenotyping plates. After pinning plates were wrapped in plastic film to prevent excessive drying and incubated at 32 °C (except for plates used in temperature assay, which were incubated at the appropriate temperature) for 20 hr prior to scanning. For conditions where the growth was highly impaired plates were incubated for 40 hr prior to scanning.

Plates were scanned and processed using *pyphe* (*Kamrad et al., 2020*; *Kamrad et al., 2022*). After processing with grid correction, median fitness and corrected (Benjamini-Hochberg) p-value thresholds (5% median fitness difference with respect to wt and adjusted p-value <0.05) were used to determine phenotypes in benign conditions (YES and EMM at 32 °C). Every mutant was measured under these conditions in over 10 repeats. To determine phenotypes in the other conditions, measured in only 4 repeats, we applied a higher threshold (10% median fitness difference with respect to wt, and growth in control media) without using p-values.

## Clustering of phenotyping data

To cluster the phenotypic data, we simplified the dataset by converting the data into ternary encoding using a median phenotypic value threshold: phenotypes showing a reduction of ≥10% on the phenotypic score were coded as –1, those showing an increase of ≥10% were coded as +1, and the weaker phenotypes in between were coded as 0. We removed the strains that contained >50% missing values and the conditions with >90% of the scores being 0.

Clusters for *Figure 3* were determined using k medoids in R (pam function from package cluster) where the distance between the genes was calculated as 1-Pearson correlation, using 8 as the optimal number of clusters determined by the R package factoextra (fviz_nbclust function).

For the networks in *Figure 6*, we used Pearson correlations and filtered on absolute r values >0.7 and adjusted p<0.01 (*Rodriguez-Lopez et al., 2022*). The network was visualized using Cytoscape and clustering was done using community clustering (GLay) from the clustermaker extension (*Morris et al., 2011*).

## NET-FF integrated method for prediction of GO molecular function and biological process terms

We developed a new computational pipeline (NET-FF) combining two independent approaches (NetHom and CATHPredictGO), for predicting GO terms associated with *S. pombe* proteins. NetHom is a machine learning method that combines protein family data from CATH functional families with protein network features generated using the DeepNF method (*Gligorijevic et al., 2018*).

CATHPredictGO is a simple decision tree approach that only uses CATH protein family data and has been endorsed for precision in multiple rounds of CAFA independent assessment (*Zhou et al., 2019a*; *Radivojac et al., 2013*; *Jiang et al., 2016*). It is included to increase confidence in predicted GO terms. Following benchmarking on various CAFA datasets, we excluded Cellular Component predictions as our tool was underperforming when predicting CC terms.

### NetHom – a machine learning predictor based on protein network and homology-based features

NetHom combines network features derived using the DeepNF approach pioneered by the Bonneau group with homology-based features from CATH-FunFams (*Sillitoe et al., 2021*). Each type of feature is described below.

### Network embeddings

Unsupervised feature learning was used to extract the context *S. pombe* genes within networks from the STRING database (v11.0) (*Szklarczyk et al., 2019*). We applied a modified version of deepNF

(*Gligorijevic et al., 2018*), that uses a multimodal deep autoencoder, a type of neural network, to embed genes in a low-dimensional space, according to their multi-network context. Individual networks were constructed for the 5100 *S. pombe* proteins contained in STRING and each of six interaction types: 'neighborhood,' 'fusion,' 'cooccurrence,' 'coexpression,' 'experimental,' and 'database.' The autoencoder compresses 30,600 dimensions of network data into 256 dimensions that can be used as features to train off-the-shelf machine learning models. A multimodal deep autoencoder was used with hidden layers for each of the six networks, followed by a single, shared embedding layer, followed by hidden layers for each of the six networks, with all layers containing 256 neurons. Sigmoid activations were used on the embedding layer, so that embedding values ranged between –1 and +1, whilst ReLU activations were used on the hidden layers. Models were trained using the Adam optimizer for 500 epochs with batch sizes of 128 examples using data from 90% of proteins. The remaining 10% of proteins were used as a validation set to monitor training using the binary cross-entropy loss function. After training, weights from the epoch with the lowest validation loss were used to generate embeddings for all proteins. Models were implemented using Keras (v2.1.5) and TensorFlow (v1.8.0).

## Homology assignments from functional families (FunFams)

To predict GO terms using homology data, we scanned the *S. pombe* proteins against Hidden Markov Models for CATH-FunFams (*Sillitoe et al., 2021*). FunFams are subsets of protein domains within a CATH Superfamily predicted to share the same function (*The Gene Ontology Consortium, 2019*; *The UniProt Consortium, 2019*). FunFams are generated using a purely sequence-based approach that segregates sets of homologous domain sequences within a CATH superfamily according to differentially conserved residue positions likely to be associated with functional properties. Our protocol uses agglomerative clustering and exploits HHsuite (*Steinegger et al., 2019*) to iteratively compare HMMs derived for clusters of relatives within a CATH superfamily. The starting clusters are sets of relatives sharing at least 90% sequence identity. At each step, the two most similar clusters are merged to ultimately give a tree of relationships across the superfamily. Subsequently, the FunFamer algorithm (*Das et al., 2015*) is used to determine whether nodes within the tree should be merged into the same functional family (FunFam) based on sharing highly conserved residues, differentially conserved in other FunFams. Each FunFam contains at least one experimentally characterized relative with an assigned GO MF or GO BP term. All 5,396 protein sequences (downloaded from PomBase on Feb 21, 2019) were searched against the FunFam hidden Markov model (HMM) library from CATH (v4.2) using hmmsearch from HMMER3 (*Das et al., 2015*; *Mistry et al., 2013*) and an E-value threshold of $E<10^{-3}$. In total, 3319 sequences had 149,099 hits to 23,900 FunFams. For machine learning, E-values were $-\log_{10}$-transformed to convert to a linear scale and transformed to a [n proteins x m FunFams] matrix, with 99.8% sparsity. For each gene and GO term pair, the corresponding position in the target matrix was set to 1 if the gene was annotated with the term. Because 23,901 FunFams were hit, the FunFam feature matrix was very wide, from a machine learning perspective. So, when predicting some GO term *g*, models were trained on FunFams that contain at least one protein annotated with *g*. GO annotations (*The Gene Ontology Consortium, 2019*; *Ashburner et al., 2000*) for all UniProt (*The UniProt Consortium, 2019*) accessions contained in FunFams were downloaded in February 2020. All annotations were included, except those with NAS, ND, TAS, or IEA evidence codes (except UniProtKB-kw IEA curated terms, which were included). GO terms were associated with FunFams by identifying all terms annotated to proteins in each FunFam. Ancestor terms that have 'is_a,' 'has_part,' 'part_of,' and 'regulates' relationships were also included. Release '2018-11-12' of the GO was used.

## Integrating the multiple data types (protein network, protein family data) using machine learning NetHom

Supervised machine learning was used to predict GO annotations for *S. pombe* proteins using random forests. We used Julia (v1.5) and the DecisionTree.jl package, which implements random forests of classification and regression trees. The cost function employed the following criteria: no maximum depth, so trees could grow arbitrarily deep; a minimum of two samples is needed to split a node, resulting in terminal nodes with single samples in each; and no minimum purity increase, here defined according to minimizing entropy. Forests of 500 trees were grown, where the trees were not pruned after they were grown. Overall model performance was estimated using fivefold stratified cross-validation. The data was shuffled before each cross-validation. Five independent trials of

cross-validation were performed to estimate the model performance under different train-test splits. Terms were predicted using the one-vs-rest multiclass strategy. Two hyperparameters, the number of features {10, 25, 50, √n} and the partial sampling of examples {0.50, 0.75, 1.00}, were optimized using an exhaustive grid search. Each combination of parameters was assessed using a nested fivefold stratified cross-validation, evaluated using the area under the precision-recall curve (AUPR). Combinations of features—network embeddings, FunFams — were benchmarked for their ability to predict *S. pombe* protein function. We also assessed the benefit of including features on growth phenotypes. Functions were predicted for *S. pombe* proteins for which growth phenotypes were collected. The 53 *S. pombe* GO Slim terms were used as targets (accessed from PomBase on Nov 14, 2018). GO Slim terms were chosen as targets, as these are sufficiently informative terms in the ontology. Following this, GO term annotations were predicted for *S. pombe* proteins using the combination of features that produced the highest AUPR. We used a fivefold stratified cross-validation strategy for prediction, whereby models predict labels for samples that they have never seen before, which forces the model to not overfit on the training data. Because we did not use growth phenotype features to make final predictions, we did not exclude any genes from our dataset. GO terms were included in the target set if they were annotated to between 50 and 1000 proteins. GO annotations were propagated to their less specific parent terms in the ontology.

## Performance of NetHom predictor

The protein network embeddings and/or CATH homology data were used as input to train a Random Forest method using one-vs-rest classification with 53 GO Slim BP terms for *S. pombe* as classes for the model (*Harris et al., 2021*). Different combinations of protein network and homology features were used to predict GO slim terms and their performance was evaluated using fivefold classification. Alone, network embeddings were the best set of features followed by FunFam homology data, with areas under the precision-recall curve (AUPR) of 0.58 and 0.42, respectively (*Figure 4—figure supplement 2*). We tested three combinations of features. The combination of network embeddings and FunFam data showed a performance that was 3% higher than the network embeddings alone, with an AUPR of 0.60 (*Figure 4—figure supplement 2*). We also tested for any benefit of using the colony-growth phenotype data described above, by including these data as a simple vector of phenotype presence or absence. Surprisingly, adding the phenotype data to the network data produced a slight drop in performance. Although the phenotype data has predictive value (*Figure 2—figure supplement 2*), these results suggest that it does not add to the comprehensive information derived from the STRING protein-interaction network. For this reason, we did not use the phenotype data for the prediction model but used it as an orthogonal cross-validation dataset.

## CATHPredictGO - protein function prediction by protein family data alone

We also implemented a protein function prediction method based on the membership of protein families. The predictor was constructed as a fast and frugal tree, which is a type of decision tree that asks a total of *n* questions and has n+1 exits. First, a protein is scanned against the FunFams from CATH (v4.3) (*Sillitoe et al., 2021*), using the FunFam inclusion thresholds. The set of non-overlapping domain hits with the highest bit scores were resolved using CATH-Resolve-Hits (*Lewis et al., 2019*). If the protein had a significant match to one or more FunFams, the FunFam match with the highest bit score above the inclusion threshold was used. All GO terms annotated to existing members of that FunFam were predicted for the protein, with probability set to the proportion of members annotated with each GO term. Low probability predictions with p<0.1 were removed. If the protein did not have a significant match to at least one FunFam, then the protein was scanned against Pfam-FunFams that we generated from Pfam v32 (*El-Gebali et al., 2019*) using the FunFam generation pipeline (*Adeyelu et al., 2023*). GO terms were predicted using Pfam-FunFams in the same way as for CATH-FunFams. GO term annotations for UniProt accessions were downloaded in Feb 2020 and associated with FunFams and Pfam-FunFams. GO terms were up-propagated to their less specific parent terms in the ontology.

## NET-FF protein function prediction by combining machine learning (NetHom) and CATHPredictGO predictions

Predicted protein functions from the machine learning and protein family-based methods were combined into a single dataset. If both methods predicted a particular GO term, with probabilities $p_1$ and p2, the combined probability was reported as $\min(p_1 + p_2, 1)$. GO taxon constraints were applied to remove GO terms that never occur in *S. pombe* taxa. GO term information content was added to each predicted GO term using the *S. pombe* GO annotation set (**Mistry and Pavlidis, 2008**). New terms that had not previously been annotated to *S. pombe* proteins were assigned the highest information content observed in the PomBase annotation set. Although both NetHom and CATHPredictGO exploit CATH FunFams they do so in different ways. Furthermore, since FunFams have been endorsed for precision in CAFA, we decided to upweight their contribution in this way.

## Pairwise GO semantic similarity calculations

Gene Ontology terms from the combined machine-learning and protein family-based methods were filtered by Information Content (IC >5) and confidence score assigned by the predictor (p>0.7), reducing the number of genes with GO terms to 2167. All available genes and their GO terms predictions were collected from PomBase (n genes = 5396). For both datasets, we used GOGO (**Zhao and Wang, 2018**) in an all-vs-all fashion to calculate the pairwise GO semantic similarity (GOGO score) for each pair of genes, i.e., genes from the NET-FF dataset (n=2167, nGO = 22,060) with a GO probability of 0.7 and information content over 5, and all genes from PomBase with an associated set of GO terms (n=5396, nGO = 45,542), resulting in 14 million and 1.5 million GOGO scores for PomBase and NET-FF, respectively. Results were subsequently separated by their ontology (Molecular Function; MFO) and (Biological Process; BPO) and deduplicated. All 5396 genes in PomBase had GOGO scores for Biological Process, while 6 genes had missing scores for Molecular Function. The majority of genes in the NET-FF dataset had resulting GOGO scores, with 1674 genes having at least one positive GO:GO score for Biological Process and 1438 genes with a positive score for Molecular Function.

## Information content calculation

The Information Content for GO terms was calculated as the negative log probability of the term occurring in all GO terms prediction for PomBase:

$$IC(GOterm) = -log(p(GOterm))$$

Where the frequency p(GOterm) is:

$$p(GO) = \frac{n_{GO'}}{N} \left| GO' \varepsilon \left\{ GO, \ GO \ descendants \right\} \right|$$

$nGO'$ is the number of annotations with the term GO′ and $N$ is the total number of GO terms in PomBase. The code to generate the IC content was adapted from the following Github gist: https://gist.github.com/avrilcoghlan/047a086c3b1b97071e177af6f0f1916d.

## Chronological lifespan measurements for experimental validation

We used our recently developed high-throughput assay (**Romila et al., 2021**) to determine the CLS of the strains in **Figure 6**. In brief, aliquots of aging cultures were taken daily and serially diluted using an automated multichannel pipette (Integra Assist; Integra Biosciences Ltd). The serially diluted droplets were pinned in quadruplicate (384-well format) onto YES agar plates using a Singer RoToR HDA pinning robot (Singer Instruments). The plates were incubated at 32 °C for 2–4 days until colonies were clearly visible. To collect images of agar plates, *pyphe-scan* was used with a transmission mode Epson V700 scanner (**Kamrad et al., 2020**; **Rodriguez-Lopez et al., 2022**). The R package *DeadOrAlive* was used to analyze the plate pictures and calculate the total number of CFUs in the aging cultures (**Romila et al., 2021**). The maximum CLS for each mutant was determined as the number of days from reaching the stationary phase (100% viability) until the cells appeared to reach 0% viability (CFUs <1). For this experiment, three independent biological repeats were measured for each strain, except for the wild-type control and *SPCC4B3.06c* mutant which were measured in two independent repeats.

## Acknowledgements

We thank Val Wood, Manuel Lera-Ramirez, and Melania D'Angiolo for helpful comments on the manuscript, Manuel Lera-Ramirez for generating genotype-to-phenotype datasets in phaf format and phenotype curation in PomBase, John Shawe-Taylor for advice on machine-learning approaches, and St John Townsend for exploring alternate clustering approaches. We dedicate this paper to the memory of St John Townsend who has passed away far too soon. This work was funded by BBSRC Research Grant BB/R009597/1 to CO and JB, and a Scholarship from the Newton-Mosharafa Fund to SH.

## Additional information

### Funding

| Funder | Grant reference number | Author |
|---|---|---|
| Biotechnology and Biological Sciences Research Council | BB/R009597/1 | María Rodríguez-López |
| Newton-Moshara Fund | Scholarship | Shaimaa Hassan |
| Biotechnology and Biological Sciences Research Council | BB/R009597/1 | Nicola Bordin |

The funders had no role in study design, data collection and interpretation, or the decision to submit the work for publication.

### Author contributions

María Rodríguez-López, Conceptualization, Resources, Data curation, Formal analysis, Supervision, Validation, Investigation, Visualization, Methodology, Writing – original draft, Writing – review and editing; Nicola Bordin, Conceptualization, Resources, Data curation, Software, Formal analysis, Investigation, Visualization, Methodology, Writing – original draft, Writing – review and editing; Jon Lees, Conceptualization, Software, Formal analysis, Supervision, Visualization, Methodology, Writing – review and editing; Harry Scholes, Software, Formal analysis, Visualization, Methodology; Shaimaa Hassan, Validation, Investigation, Methodology; Quentin Saintain, Validation, Methodology; Stephan Kamrad, Conceptualization, Methodology, Writing – review and editing; Christine Orengo, Conceptualization, Supervision, Funding acquisition, Investigation, Writing – original draft, Project administration, Writing – review and editing; Jürg Bähler, Conceptualization, Data curation, Supervision, Funding acquisition, Investigation, Writing – original draft, Project administration, Writing – review and editing

### Author ORCIDs

María Rodríguez-López ⓘ https://orcid.org/0000-0002-2066-0589
Nicola Bordin ⓘ http://orcid.org/0000-0002-6568-9035
Jon Lees ⓘ http://orcid.org/0000-0003-3925-1424
Christine Orengo ⓘ http://orcid.org/0000-0002-7141-8936
Jürg Bähler ⓘ http://orcid.org/0000-0003-4036-1532

Reviewer #1 (Public Review): https://doi.org/10.7554/eLife.88229.3.sa1
Reviewer #2 (Public Review): https://doi.org/10.7554/eLife.88229.3.sa2
Reviewer #3 (Public Review): https://doi.org/10.7554/eLife.88229.3.sa3
Author Response https://doi.org/10.7554/eLife.88229.3.sa4

## Additional files

### Supplementary files

• Supplementary file 1. Supporting data for phenomics assays.

- Supplementary file 2. Data for NET-FF predictions and integrated analyses.
- MDAR checklist

## Data availability

Data generated or analysed during this study are included in the manuscript and supporting files. The code essential to generate these findings is freely available on GitHub (copy archived at *Rodríguez-López et al., 2023*). The code is written in Julia and Python3. To generate genotype-to-phenotype datasets in phaf format that can be directly submitted to PomBase, we used code available on Zenodo.

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
