## [Editor Report · eLife assessment]

This **important** study combines extensive phenotyping of genome-wide deletion mutants and machine learning-based prediction to generate a large scale resource for understanding the functions of thousands of fission yeast protein-coding genes. This resource is supported by **convincing** phenotyping data and state-of-the-art bioinformatic analyses and will be of interest to many geneticists.

---

## [Referee Report · Reviewer #1 (Public Review)]

In this manuscript, the authors aimed to provide information about the likely function of uncharacterised genes in fission yeast. The authors highlight the bias in the literature to well-studied genes/proteins and the fact that the functions of many proteins that are conserved from yeast to humans remain unknown. Initial functional characterisation could provide the impetus for researchers to dedicate time and resources to detailed investigations of protein function. The authors subject the fission yeast deletion set to a battery of perturbations (drug treatments etc) and measured the resultant colony size. In total, 131 conditions were analysed for nearly 3,500 mutants, representing a rich dataset. Clustering analysis was then used to identify common phenotype patterns and thereby infer protein functions using a "guilt by association approach. To assign potential GO terms to uncharacterised proteins, the authors developed a new computational approach (NET-FF) which combined two previous approaches, which they validated against curated annotations on the *S. pombe* database Pombase. Finally, the authors chose a group of genes which their analysis predicted to be involved in cellular ageing for experimental validation, cross-validating a priority unstudied novel gene (SPAC23C4.09c) to be involved in this process. Overall, the functional analysis performed in this manuscript is rigorous, thorough and incorporates some novel approaches leading to new insights and predicted protein functions. It will be an important resource for the fission yeast community.

---

## [Referee Report · Reviewer #2 (Public Review)]

This manuscript describes colony-growth phenotypes to measure the fitness of deletion mutants for 3509 non-essential *S. pombe* genes in 131 conditions. 3492 mutants, including 124 mutants of 'priority unstudied' proteins conserved in humans, providing varied functional clues.

Phenotype-correlation networks provide evidence for the roles of poorly characterized proteins through guilt by association with known proteins. Gene Ontology (GO) terms were predicted using machine learning methods that take advantage of protein-network and protein-homology data.

Integrated analyses produced 1,675 novel GO predictions for 783 genes, including 47 predictions for 23 priority unstudied proteins. Experimental validation for genes involved in cellular ageing were obtained.

A method called NET-FF, which combines network embeddings and protein homology data to predict GO annotations, was developed. The authors demonstrate NET-FF predicts GO terms better than random and compare the information content of the predicted terms with the PomBase GO annotations. The phenotypic data was used to filter the GO annotation predictions made by NET-FF and then explore specific biological examples supported by both datasets

This is a very impressive and rich resource of phenotypic data and it will be particularly useful for the *S. pombe* research community and generally useful for the functional characterization of highly conserved eukaryotic genes. Overall, the analysis is powerful and sound.

---

## [Referee Report · Reviewer #3 (Public Review)]

Fission yeast is an important model organism and studies on fission yeast have provided many key insights into the understanding of genes and biological pathways. However, even in such a well-studied model organism, there are still many genes without known functions.

In this work, the authors took advantage of the availability of genome-wide fission yeast deletion mutants to systematically analyze the mutant phenotypes under 131 different conditions. This effort generated a genotype-phenotype dataset larger than the currently curated genotype-phenotype dataset, which is derived from studies over many decades by hundreds of fission yeast laboratories. The authors used the dataset to construct gene clusters that provide functional clues for many genes without previously known functions, including ones conserved in humans. This rich resource will surely be highly useful to the fission yeast community and beyond.

In addition, the authors also used machine learning to generate functional predictions of fission yeast genes and yield novel understandings, which are validated by experimental analysis of new ageing-related genes.

Overall, this study provides unprecedented and highly valuable resources for understanding fission yeast gene functions.

---

## [Author Response]

The following is the authors’ response to the original reviews.

We thank the three reviewers for their positive comments and helpful suggestions. We have addressed the issues raised which have helped to improve the manuscript. Below, we address the specific points with detailed responses.

**Reviewer #1 (Recommendations For The Authors):**
Minor comments1. Figure 2 - figure supplement 1. The figure states minimal medium while the legend states rich medium.

We have corrected the legend as the experiment was done in minimal medium.

2. Figure 3B - the statements in the text do not seem to match what is in the figure. "Cluster 1 (293 genes, 12 priority unstudied) is enriched for genes showing high expression variability across different conditions (71) and for genes induced during meiotic differentiation (72) and in response to TORC1 inhibitors (29). Cluster 2 (570 genes, 20 priority unstudied) is enriched for phenotypes related to cell mating and sporulation, e.g. 'incomplete cell-wall disassembly at cell fusion site' or 'abnormal shmoo morphology'". These terms high expression variability, meiotic differentiation, TORC1 inhibitors, cell mating and sporulation/abnormal shmoo morphology" are not seen in the figure.

As stated in the Results, we have carried out analyses with both Metascape and AnGeLi for functional enrichments in different GO and KEGG pathway terms (Figure 3B; Metascape) and/or among genes from published expression or phenotyping studies (AnGeLi). The enrichments for expression variability, meiotic differentiation, TORC1 inhibitors, and cell mating/sporulation/abnormal shmoo morphology are not based on GO terms but on lists from published expression and phenotyping experiments. We have slightly edited the sentence in the Results to make this clearer.

3. The authors could consider citing a systematic screen for sporulation in the introduction (PMID: 292590)

We have cited 17 papers for growth screens under different conditions using similar approaches as used by us. Given that we already cite 100 papers, we did not choose to cite numerous other papers reporting screens for more complex phenotypes (cell morphology, mating, meiosis, recombination, etc), which are not directly relevant to our study here.

Reference PMID: 292590 refers to a 1979 paper in the German Dentist Journal.

**Reviewer #2 (Recommendations For The Authors):**
General comments1. The authors use their NET-FF approach to predict GO Biological Process and Molecular Function terms (Figure 4). Why was the Cellular Component ontology not included? In general, gene and protein functional characterization is best described by the Biological Process and Cellular Component ontologies, whereas Molecular Function describes the biochemical activity of a protein. In other words, proteins which share Biological Process and/or Cellular Component annotations often function in the same module, which may not be the case for shared Molecular Function annotations.

We did not include Cellular Component because in previous benchmarking of our method using CAFA datasets our approach did not perform well at predicting Cellular Component. This aspect is harder to pick up from homology data and protein network data and is generally the toughest challenge in CAFA. In contrast, our predictions of Biological Process and Molecular Function are competitive with other methods. We have now made the reason for omitting Cellular Component clearer in the Methods.

2. The authors use protein embeddings produced by integrating 6 STRING networks using the deepNF method. One of these networks is the "database" network. According to STRING (https://academic.oup.com/nar/article/47/D1/D607/5198476): "The database channel is based on manually curated interaction records assembled by expert curators, at KEGG, Reactome, BioCyc and Gene Ontology, as well as legacy datasets from PID and BioCarta". If one of the input networks contains information from GO, and then embeddings containing this information are used to predict GO annotations, are the authors not then leaking annotations which could improve downstream GO annotation predictions? It would be valuable to demonstrate to what extent the "database" network is contributing by repeating the GO prediction analyses with this network removed.

We agree and also pointed out this circularity in the manuscript. We used an independent dataset – phenotype data – to benchmark our method, which showed good performance. Note that this study did not aim to develop a completely new method or improve on deepNF and CATH-FunFams but to integrate and exploit their combined power. For that reason, we wanted to keep as many high-quality curated edges in the STRING network as possible. Combining these independent methods brings synergies from their complementary approaches to facilitate interpretation of gene function.

Minor comments1. Ternary encoding was used as a preprocessing step on the phenotype data before clustering was performed. An explanation of why this encoding was necessary (as opposed to a normalization/standardization approach) would be helpful.

Ternary encoding was not strictly necessary but provided more nuanced and coherent clusters. Some conditions and mutants were associated with much larger phenotypic responses which disproportionately influenced the clustering. After trying different approaches, we followed the recommendations from the R package microbialPhenotypes (https://github.com/peterwu19881230/microbialPhenotypes), which is now specified in the legend of Fig. 3A. Discretizing the data also helped to compare phenotypes across different types of mutants, and we have applied this approach previously in our phenomics study of non-coding RNA mutants (Rodriguez-Lopez et al. eLife 2022). Moreover, this approach allowed us to generate vectors of phenotypes for calculating phenotypic distances between mutants (including hamming distance or Pearson correlations), which supported the posterior cluster analysis using Cytoscape.

2. The authors use a validation set to perform early-stopping on the deepNF model. However, it appears that the validation set proteins are then used in downstream analyses anyway: "After training, weights from the epoch with the lowest validation loss were used to generate embeddings for all proteins" (my emphasis). In the case where the model was being used to generalize to new proteins (such as classification), this analysis would not be a valid way to perform hyperparameter tuning (e.g. early-stopping) since the validation set is then used in downstream analyses. However, deepNF is performing an unsupervised, multi- network encoding on all the available datapoints (proteins). In the case where only deepNF loss is being used to tune the hyperparameters, it's not necessary to use a held-out validation set - it is appropriate to use the full set of proteins to do this.

Our Random Forest consisted of 500 trees with default values for the number of sub- features as √n and partial sampling of 0.7. GO terms were predicted using 5-fold cross- validation. Changing parameters showed that our model was robust to the values of the hyperparameters, so we settled on our initial model.

3. The NET-FF hyperparameter tuning results should be made available in the supplement.

We do not think this would be useful for the reason described in the reply above.

**Reviewer #3 (Recommendations For The Authors):**
Major points1. Why were the quantitive colony size data converted to -1, 0, and 1?It is unclear to me why the authors decided to convert the colony size data to ternary encoding of -1, 0, and 1. The original colony size data seem to be of fairly high precision so that the authors can detect a 5% difference from the wild type. I guess the authors must have tried using the quantitive colony size data for clustering analysis and found the results unsatisfactory. If that is the case, can the authors provide some possible explanations?

A similar query has been raised by Reviewer 2. Ternary encoding provided more nuanced and coherent clusters. Some conditions and mutants were associated with much larger phenotypic responses which disproportionately influenced the clustering. After trying different approaches, we followed the recommendations from the R package microbialPhenotypes, as now specified in the legend of Fig. 3A. Discretizing the data also helped to compare phenotypes across different types of mutants, and we have applied this approach previously in our phenomics study of non-coding RNA mutants (Rodriguez-Lopez et al. eLife 2022). Moreover, this approach allowed us to generate vectors of phenotypes for calculating phenotypic distances between mutants (including hamming distance or Pearson correlations), which supported the posterior cluster analysis using Cytoscape.

2. What do 5% difference and 10% difference look like?The authors used 5% difference and 10% difference as cutoffs. I am curious whether a 5% difference in colony size is obvious to human eyes. Can the authors show some plate images and label colonies that differ from the wild type by about 5% and 10%? It will help readers understand the thresholds used for determining whether a mutant has a phenotype.

Showing the original ‘raw’ colonies would not be meaningful because all colony sizes have been grid-corrected as described (Kamrad et al. eLife 2020). The grid correction takes care of three issues: (1) it converts colony size into an easily interpretable value by reporting a ratio relative to wild type; (2) it makes results comparable across different plates/batches; and (3) it corrects for within-plate positional effects which become apparent due to the same wild-type grid strain showing different fitness in different plate positions. But in principle, detecting a 5% difference in colony size by eye would be hard, and multiple measurements are required (>10 repeats) to obtain statistically reliable results. Author response image 1 shows the grid colonies in red frames and numbers at bottom right of colonies indicate the corrected effect sizes. Colony 17-8 (top right) is an example of a colony differing by 5% compared to neighbouring colonies 16-8 and 17-9.

**Author response image 1. sa4fig1:** 

3. How were the phenotyping conditions chosen?I am sure that the authors have put a lot of thoughts into designing the 131 phenotyping conditions. It will benefit the readers if the authors can explain how these conditions were chosen. For example, what literature precedents were considered and which conditions have never been examined before in *S. pombe* research? For drug treatment conditions, werepilot tests done to choose drug doses based on the growth inhibition effects on the wild type?

We have used a wide range of different types of conditions that affect diverse processes (see colour legend on top of Fig. 3A). This was based on our previous experience and selection of conditions in large-scale phenotyping of wild strains (Jeffares et al. Nature Genetics 2015) and non-coding RNA mutants (Rodriguez-Lopez et al. eLife 2022). For previously applied conditions (e.g. oxidants), we used literature precedents for the doses, while for other conditions, we used trial and error to adjust the diose such that wild-type cell growth is barely inhibited. For some drugs and stresses, we assayed both low and high doses, in which wild-type cell growth is normal or inhibited, respectively, to uncover both sensitive or resistant mutants.

Minor points1. One of the growth condition is "YES_ethanol_1percent_no_glucose". I am curious how this is possible, as *S. pombe* cannot use ethanol as a carbon source.

We assume that the cells contain sufficient internal glucose to fuel growth and division for a few cycles before running out of glucose. Thus, cells showed some residual growth on this medium, but growth is indeed very limited. Nevertheless, we could identify both sensitive and resistant mutants in this condition.

2. Abstract "over 900 new proteins affected the resistance to oxidative stress". This sentence should be rephrased. Perhaps it is better to say "over 900 proteins were newly implicated in the resistance to oxidative stress".

Yes, we have edited the sentence as suggested.

3. Page 4 "*S. pombe* encodes 641 'unknown' genes (PomBase, status March 2023). " "Among these 643 unknown proteins, many are apparently found only in the fission yeast clade, but 380 are more widely conserved. " Which number is correct, 641 or 643?

These numbers keep changing slightly. We now consistently use 641, the number from March 2023.

4. Page 4 "These priority unstudied proteins have not been directly studied in any organism but can be assumed to have pertinent biological roles conserved over 500 million years of evolution. " According to http://timetree.org/, *S. pombe* and H. sapiens diverged about 1275 million years ago.

We have now changed ‘over 500 million’ to ‘over 1000 million’, although there are of course different estimates for these times.

5. "Using these potent wet and dry methods, we obtained 103,520 quantitative phenotype datapoints for 3,492 non-essential genes across 131 diverse conditions."

I think "quantitative phenotype datapoints" are generated using wet methods, not dry methods.Yes, we have now deleted ‘Using these potent wet and dry methods,’ and start the sentence with ‘We obtained…’

6. Abstract "We assayed colony-growth phenotypes to measure the fitness of deletion mutants for all 3509 non-essential genes"Page 6 "We performed colony-based phenotyping of the deletion mutants for all non- essential *S. pombe* genes"It is not clear to me how the authors can claim that the 3509 non-essential genes correspond to "all non-essential S. pombe genes". The authors should explain how they classify *S. pombe* genes into essential genes and non-essential genes. The deletion project papers (Kim et al. 2010 and Hayles et al. 2013) provided binary classification for most but not all genes, as there are genes whose deletion mutants were not generated by the deletion project. PomBase does not use a binary classification and there are a number of genes deemed "Gene Deletion Viability: Depends on conditions" by PomBase.

We used the latest deletion library (Bioneer Version 5) as well as additional deletion mutants published by Kathy Gould and colleagues, which together should capture all non- essential genes. But we agree that non-essentiality is not that clear-cut and context- dependent. So we have deleted ‘all’ in the two sentences highlighted above.

7. Page 20 "Other clusters contained mostly genes involved in vacuolar/endosomal transport and peroxisome function, along with poorly characterized genes (Figure 6B)."This sentence needs rephrasing. Perhaps it is better to say "Cluster 31 and cluster 22 contained respectively mostly genes involved in vacuolar/endosomal transport and peroxisome function, along with poorly characterized genes (Figure 6B)."

We have edited this sentence to ‘Cluster 31 and Cluster 22 contained mostly genes involved in vacuolar/endosomal transport and peroxisome function, respectively, along with poorly characterized genes (Figure 6B).’

8. Legend of Figure 2-figure supplement 1A"Left: Volcano plot of mutant colony sizes for priority unstudied genes (green) and all other genes (grey) growing in rich medium. " I think "rich medium" should be "minimal medium".

Yes, we have now corrected this.